# FLOW MATCHING POLICY GRADIENTS

**David McAllister**[1*]   **Songwei Ge**[1*]   **Brent Yi**[1*]   **Chung Min Kim**[1]
**Ethan Weber**[1]   **Hongsuk Choi**[1]   **Haiwen Feng**[1,2]   **Angjoo Kanazawa**[1]

[1]UC Berkeley   [2]Max Planck Institute for Intelligent Systems

## ABSTRACT

Flow-based generative models, including diffusion models, excel at modeling continuous distributions in high-dimensional spaces. In this work, we introduce Flow Policy Optimization (FPO), a simple on-policy reinforcement learning algorithm that brings flow matching into the policy gradient framework. FPO casts policy optimization as maximizing an advantage-weighted ratio computed from the conditional flow matching loss, in a manner compatible with the popular PPO-clip framework. It sidesteps the need for exact likelihood computation while preserving the generative capabilities of flow-based models. Unlike prior approaches for diffusion-based reinforcement learning that bind training to a specific sampling method, FPO is agnostic to the choice of diffusion or flow integration at both training and inference time. We show that FPO can train diffusion-style policies from scratch in a variety of continuous control tasks. We find that flow-based models can capture multimodal action distributions when necessary and achieve comparative or higher performance than Gaussian policies, particularly in under-conditioned settings. For overview of FPO's key insights and interactive results please see our anonymized supplemental website.

## 1   INTRODUCTION

Flow-based generative models—particularly diffusion models—have emerged as powerful tools for generative modeling across the domains of images (Saharia et al., 2022; Ho et al., 2022b), videos (Brooks et al., 2024; Polyak et al., 2024; Veo-Team et al., 2024), speech (Liu et al., 2023), audio (Kong et al., 2021), robotics (Chi et al., 2024b), and molecular dynamics (Raja et al., 2025). In parallel, reinforcement learning (RL) has proven to be effective for optimizing neural networks with non-differentiable objectives, and is widely used as a post-training strategy for aligning foundation models with task-specific goals (Chu et al., 2025; Liu et al., 2024).

In this work, we introduce Flow Policy Optimization (FPO), a policy gradient algorithm for optimizing flow-based generative models. FPO reframes policy optimization as maximizing an advantage-weighted ratio computed from the conditional flow matching (CFM) objective (Lipman et al., 2023). Intuitively, FPO shapes probability flow to transform Gaussian noise into high-reward actions by reinforcing its experience using flow matching. The method is simple to implement and can be readily integrated into standard techniques for stochastic policy optimization. We use a PPO-inspired surrogate objective, enabling flow policies as a drop-in replacement for Gaussian policies in existing PPO frameworks.

FPO offers several key advantages. It sidesteps the complex likelihood calculations typically associated with flow-based models, instead using the flow matching loss as a surrogate for log-likelihood in the policy gradient. The objective increases the evidence lower bound of high-reward actions. FPO treats the sampling procedure as a black box during rollouts, which allows for flexible integration with any sampling approach—whether deterministic or stochastic, and with any number of steps during training and inference.

We theoretically analyze FPO's correctness and empirically validate its performance across a diverse set of tasks. These include a GridWorld environment, 10 continuous control tasks from MuJoCo Playground (Zakka et al., 2025), and high-dimensional humanoid control—all trained from scratch.

---

*Equal contribution.

FPO demonstrates robustness across tasks, enabling effective training of flow-based policies in high-dimensional domains. We probe flow policies learned in the toy GridWorld environment and find that on states with multiple possible optimal actions, it learns multimodal action distributions. On humanoid control tasks, we show that the expressivity of flow matching enables single-stage training of under-conditioned control policies, where only root-level commands are provided. In contrast, Gaussian policies struggle to learn viable walking behaviors in such cases, suggesting the practical benefits of more flexible policy representations. We will open-source our code to enable further research in this direction.

## 2 RELATED WORK

**Policy Gradients.** We study on-policy reinforcement learning, where a parameterized policy is optimized to maximize cumulative reward in a provided environment. This is commonly solved with policy gradient techniques, which bypass the need for differentiable environment rewards by weighting action log-probabilities with observed rewards or advantages (Sutton et al., 1999; Williams, 1992; Kakade, 2002; Peters & Schaal, 2008; Schulman et al., 2015a; 2017; Mnih et al., 2016; Wang et al., 2016; Shao et al., 2024). Policy gradient methods are central in learning policies for general continuous control tasks (Duan et al., 2016; Huang et al., 2024), robot locomotion (Rudin et al., 2022; Schwarke et al., 2023; Mittal et al., 2024; Allshire et al., 2025) and manipulation (Akkaya et al., 2019; Chen et al., 2021; Qi et al., 2023; 2025). They have also been adopted increasingly for searching through and refining prior distributions in pretrained generative models. This has proven effective for alignment with human preferences (Ouyang et al., 2022; Christiano et al., 2023) and improving reasoning using verifiable rewards (DeepSeek-AI et al., 2025).

In this work, we propose a simple algorithm for training flow-based generative policies, such as diffusion models, under the policy gradient framework. By leveraging recent insights from flow matching (Lipman et al., 2023), we train policies that can represent richer distributions than the diagonal Gaussians that are most frequently used for reinforcement learning for continuous control (Rudin et al., 2022; Schwarke et al., 2023; Mittal et al., 2024; Allshire et al., 2025; Qi et al., 2023; 2025), while remaining compatible with standard actor-critic training techniques.

**Diffusion Models.** Diffusion models are powerful tools for modeling complex continuous distributions and have achieved remarkable success across a wide range of domains. These models have become the predominant approach for generating images (Ho et al., 2020; Song et al., 2022; Rombach et al., 2022; Song & Ermon, 2020), videos (Ho et al., 2022c; Singer et al., 2022; Ho et al., 2022a; Brooks et al., 2024), and more recently, robot actions (Chi et al., 2024b; Black et al., 2024; NVIDIA et al., 2025). In these applications, diffusion models aim to sample from a data distribution of interest, whether scraped from the internet or collected through human teleoperation.

Flow matching (Lipman et al., 2023) simplifies and generalizes the diffusion model framework. It learns a vector field that transports samples from a tractable prior distribution to the target data distribution. The conditional flow matching (CFM) objective trains the model to denoise data that has been perturbed with Gaussian noise. Given data $x$ and noise $\epsilon \in \mathcal{N}(0, I)$, the CFM objective can be expressed as:

$$\mathcal{L}_{\text{CFM},\theta} = \mathbb{E}_{\tau, q(x), p_\tau(x_\tau | x)} \| \hat{v}_\theta(x_\tau, \tau) - u(x_\tau, \tau \mid x) \|_2^2, \tag{1}$$

where $x_\tau = \alpha_\tau x + \sigma_\tau \epsilon$ represents the partially noised sample at flow step $\tau$, an interpolation of noise and data with a schedule defined by hyperparameters $\alpha_\tau$ and $\sigma_\tau$. $\hat{v}_\theta(x_\tau, \tau)$ is the model's estimate of the velocity to the original data, and $u(x_\tau, \tau \mid x)$ is the conditional flow $x - \epsilon$. The learned velocity field is a continuous mapping that transports samples from a simple, tractable distribution (e.g. Gaussian noise) to the training data distribution through ODE integration.

Optimizing likelihoods directly through flow models requires divergence estimation (Skreta et al., 2025) and is computationally prohibitive. Instead, flow matching optimizes variational lower bounds of the likelihood with the simple denoising loss above. In this work, we leverage flow matching directly within the policy gradient formulation.

**Diffusion Policies.** Diffusion-based policies have shown promising results in robotics and decision-making applications (Chi et al., 2024a; Ajay et al., 2023; Black et al., 2024). Most existing approaches train these models via behavior cloning (Janner et al., 2022; Chi et al., 2024b), where the policy is supervised to imitate expert trajectories without using reward feedback.

Recent work by Psenka et al. (2023) explores off-policy training of diffusion policies via Q-score matching. While off-policy reinforcement learning continues to make progress (Seo et al., 2025; Fujimoto et al., 2018), on-policy methods dominate practical applications today. Methods like DDPO Black et al. (2023), DPPO Ren et al. (2024), and Flow-GRPO Liu et al. (2025) adopt on-policy policy gradient methods by treating initial noise values as observations from the environment, framing the denoising process as a Markov decision process, and training each step as a Gaussian policy using PPO. Our approach differs by directly integrating the conditional flow matching (CFM) objective into a PPO-like framework Since FPO integrates flow matching as its fundamental primitive, it is agnostic to the choice of sampling method during both training and inference.

# 3 FLOW MATCHING POLICY GRADIENTS

## 3.1 POLICY GRADIENTS AND PPO

The goal of reinforcement learning is to learn a policy $\pi_\theta$ that maximizes expected return in a provided environment. At each iteration of online reinforcement learning, the policy is rolled out to collect batches of observation, action, and reward tuples $(o_t, a_t, r_t)$ for each environment timestep $t$. These rollouts can used in the policy gradient objective (Sutton et al., 1999) to increase likelihood of actions that result in higher rewards:

$$\max_\theta \; \mathbb{E}_{a_t \sim \pi_\theta(a_t|o_t)} \left[ \log \pi_\theta(a_t \mid o_t) \hat{A}_t \right], \tag{2}$$

where $\hat{A}_t$ is an advantage estimated from the rollout's rewards $r_t$ and a learned value function (Schulman et al., 2015b).

The vanilla policy gradient is valid only locally around the current policy parameters. Large updates can lead to policy collapse or unstable learning. To address this, PPO (Schulman et al., 2017) incorporates a trust region by clipping the likelihood ratio:

$$\max_\theta \; \mathbb{E}_{a_t \sim \pi_{\text{old}}(a_t|o_t)} \left[ \min \left( r(\theta)\hat{A}_t, \; \text{clip}(r(\theta), 1 - \varepsilon^{\text{clip}}, 1 + \varepsilon^{\text{clip}})\hat{A}_t \right) \right], \tag{3}$$

where $\varepsilon^{\text{clip}}$ is a tunable threshold and $r(\theta)$ is the ratio between current and old action likelihoods:

$$r(\theta) = \frac{\pi_\theta(a_t \mid o_t)}{\pi_{\text{old}}(a_t \mid o_t)}. \tag{4}$$

PPO is popular choice for on-policy reinforcement learning because of its stability, simplicity, and performance. Like the standard policy gradient, however, it requires exact likelihoods for sampled actions. These quantities are tractable for simple Gaussian or categorical action spaces, but computationally prohibitive to estimate for flow matching and diffusion models.

## 3.2 FLOW POLICY OPTIMIZATION

We introduce Flow Policy Optimization (FPO), an online reinforcement learning algorithm for policies represented as flow models $\hat{v}_\theta$. There are two key differences in practice from Gaussian PPO. During rollouts, a flow model transforms random noise into actions via a sequence of learned transformations, enabling much more expressive policies than those used in standard PPO. Also, to update the policy, the Gaussian likelihoods are replaced with a transformed flow matching loss.

Instead of updating exact likelihoods, we propose a proxy $\hat{r}^{\text{FPO}}$ for the likelihood ratio. FPO's overall objective is the same as Equation 3, but with the ratio substituted:

$$\max_\theta \; \mathbb{E}_{a_t \sim \pi_{\text{old}}(a_t|o_t)} \left[ \min \left( \hat{r}^{\text{FPO}}(\theta)\hat{A}_t, \; \text{clip}(\hat{r}^{\text{FPO}}(\theta), 1 - \varepsilon^{\text{clip}}, 1 + \varepsilon^{\text{clip}})\hat{A}_t \right) \right]. \tag{5}$$

Intuitively, FPO's goal is to steer the policy's probability flow toward high-return behavior. Instead of computing likelihoods, we construct a simple ratio estimate using standard flow matching losses:

$$\hat{r}^{\text{FPO}}(\theta) = \exp(\hat{\mathcal{L}}_{\text{CFM},\theta_{\text{old}}}(a_t; o_t) - \hat{\mathcal{L}}_{\text{CFM},\theta}(a_t; o_t)), \tag{6}$$

which, as we will discuss, can be derived from optimizing the evidence lower bound.

For a given action and observation pair, $\hat{\mathcal{L}}_{\text{CFM},\theta}(a_t; o_t)$ is an estimate of the per-sample conditional flow matching loss $\mathcal{L}_{\text{CFM},\theta}(a_t; o_t)$:

$$\hat{\mathcal{L}}_{\text{CFM},\theta}(a_t; o_t) = \frac{1}{N_{\text{mc}}} \sum_i^{N_{\text{mc}}} \ell_\theta(\tau_i, \epsilon_i) \tag{7}$$

$$\ell_\theta(\tau_i, \epsilon_i) = ||\hat{v}_\theta(a_t^{\tau_i}, \tau_i; o_t) - (a_t - \epsilon_i)||_2^2 \tag{8}$$

$$a_t^{\tau_i} = (1 - \tau_i)a_t + \tau_i\epsilon_i, \tag{9}$$

where we denote flow timesteps with $\tau$ and environment timesteps with $t$. We include both timesteps in $a_t^\tau$, which represents an action at rollout time $t$ with noise level $\tau$ following Eq. 1 with OT schedule (Lipman et al., 2023). $\hat{\mathcal{L}}_{\text{CFM},\theta_{\text{old}}}$ and $\hat{\mathcal{L}}_{\text{CFM},\theta}$ share $\epsilon_i \sim N(0, I)$ and $\tau_i \in [0, 1]$ samples.

**Properties.** FPO's ratio estimate in Equation 6 serves as a drop-in replacement for the PPO likelihood ratio. FPO therefore inherits compatibility with advantage estimation methods like GAE (Schulman et al., 2015b) and GRPO (Shao et al., 2024). Without loss of generality, it is also compatible with flow and diffusion implementations based on estimating noise $\epsilon$ (Ho et al., 2020) or clean action $a_t$ (Ramesh et al., 2022), which can be reweighted for mathematical equivalence to $\mathcal{L}_{\theta,\text{CFM}}$ (Karras et al., 2022).

## 3.3 FPO SURROGATE OBJECTIVE

Exact likelihood is computationally expensive even to estimate in flow-based models. Instead, it is common to optimize the evidence lower bound (ELBO) as a proxy for log-likelihood:

$$\text{ELBO}_\theta(a_t \mid o_t) = \log \pi_\theta(a_t \mid o_t) - \mathcal{D}_\theta^{\text{KL}}, \tag{10}$$

where $\mathcal{D}_\theta^{\text{KL}}$ is the KL gap between the ELBO and true log-likelihood and $\pi_\theta$ is the distribution captured by sampling from the flow model. Flow matching and diffusion models use the conditional flow matching loss, a simple denoising reconstruction objective. Prior work (Kingma et al., 2023) shows that the CFM loss (Eq. 1) corresponds to the negative ELBO. (Kingma et al., 2023). Using this fact, the FPO ratio in Eq. 6 corresponds to the ratio of ELBOs under current and old policies:

$$r^{\text{FPO}}(\theta) = \frac{\exp(\text{ELBO}_\theta(a_t \mid o_t))}{\exp(\text{ELBO}_{\theta_{\text{old}}}(a_t \mid o_t))}. \tag{11}$$

Decomposing this ratio reveals a scaled variant of the true likelihood ratio (Equation 4):

$$r^{\text{FPO}}(\theta) = \underbrace{\frac{\pi_\theta(a_t \mid o_t)}{\pi_{\theta_{\text{old}}}(a_t \mid o_t)}}_{\text{Likelihood}} \underbrace{\frac{\exp(\mathcal{D}_{\theta_{\text{old}}}^{\text{KL}})}{\exp(\mathcal{D}_\theta^{\text{KL}})}}_{\text{Inv. KL Gap}}. \tag{12}$$

Here, the ratio decomposes into the standard likelihood ratio and an inverse correction term involving the KL gap. Maximizing this ratio therefore increases the modeled likelihood while reducing the KL gap—both of which are beneficial for policy optimization. The former encourages the policy to favor actions with positive advantage, while the latter tightens the approximation to the true log-likelihood.

## 3.4 ESTIMATING THE FPO RATIO WITH FLOW MATCHING

We estimate the FPO ratio using the flow matching objective directly, which follows from the relationship between the weighted denoising loss $\mathcal{L}_\theta^w$ and the ELBO established by Kingma & Gao (2023). $\mathcal{L}_\theta^w$ is a more general form of the flow matching and denoising diffusion loss that parameterizes the model as predicting $\hat{\epsilon}_\theta$, an estimate of the true noise $\epsilon$ present in the model input.

The weighted denoising loss $\mathcal{L}_\theta^w$ for a clean action $a_t$ takes the form:

$$\mathcal{L}_\theta^w(a_t) = \frac{1}{2}\mathbb{E}_{\tau \sim \mathcal{U}(0,1), \epsilon \sim \mathcal{N}(0,I)} \left[ w(\lambda_\tau) \cdot \left( -\frac{d\lambda}{d\tau} \right) \cdot \|\hat{\epsilon}_\theta(a_t^\tau; \lambda_\tau) - \epsilon\|_2^2 \right], \tag{13}$$

where $w$ is a choice of weighting and $\lambda_\tau$ represents the log-SNR at noise level $\tau$. We estimate this value with Monte Carlo draws of timestep $\tau$ and noise $\epsilon$:

$$\ell_\theta^w(\tau, \epsilon) = \frac{1}{2}w(\lambda_\tau) \cdot \left( -\frac{d\lambda}{d\tau} \right) \cdot \|\hat{\epsilon}_\theta(a_t^\tau; \lambda_\tau) - \epsilon\|_2^2. \tag{14}$$

---

**Algorithm 1** Flow Policy Optimization (FPO)

---

**Require:** Policy parameters $\theta$, value function parameters $\phi$, clip parameter $\epsilon$, MC samples $N_{\mathrm{mc}}$
1: **while** not converged **do**
2:   Collect trajectories using any flow model sampler and compute advantages $\hat{A}_t$
3:   For each action, store $N_{\mathrm{mc}}$ timestep-noise pairs $\{(\tau_i, \epsilon_i)\}$ and compute $\ell_\theta(\tau_i, \epsilon_i)$
4:   $\theta_{\mathrm{old}} \leftarrow \theta$
5:   **for** each optimization epoch **do**
6:     Sample mini-batch from collected trajectories
7:     **for** each state-action pair $(o_t, a_t)$ and corresponding MC samples $\{(\tau_i, \epsilon_i)\}$ **do**
8:       Compute $\ell_\theta(\tau_i, \epsilon_i)$ using stored $(\tau_i, \epsilon_i)$
9:       $\hat{r}_\theta \leftarrow \exp\left(-\frac{1}{N_{\mathrm{mc}}} \sum_{i=1}^{N_{\mathrm{mc}}} (\ell_\theta(\tau_i, \epsilon_i) - \ell_{\theta_{\mathrm{old}}}(\tau_i, \epsilon_i))\right)$
10:       $L^{\mathrm{FPO}}(\theta) \leftarrow \min(\hat{r}_\theta \hat{A}_t, \mathrm{clip}(\hat{r}_\theta, 1 \pm \epsilon)\hat{A}_t)$
11:     **end for**
12:     $\theta \leftarrow \mathrm{Optimizer}(\theta, \nabla_\theta \sum L^{\mathrm{FPO}}(\theta))$
13:   **end for**
14:   Update value function parameters $\phi$ like standard PPO
15: **end while**

---

The choice of weighting $w$ incorporates the conditional flow matching loss and standard diffusion loss as specific cases of a more general family $\mathcal{L}_\theta^w(a_t)$.

We focus here on the constant weight case $w(\lambda_\tau) = 1$ (diffusion schedule), which yields the simplest theoretical connection. Similar results hold for many popular schedules, including optimal transport (Lipman et al., 2023) and variance preserving schedules. Please see the supplementary material for details.

For the diffusion schedule, Kingma & Gao (2023) proves that:

$$\mathcal{L}_\theta^w(a_t) = -\mathrm{ELBO}_\theta(a_t) + c, \tag{15}$$

where $c$ is a constant w.r.t $\theta$. Geometrically, minimizing $\mathcal{L}_\theta^w(a_t)$ points the flow more toward $a_t$. Minimizing $\mathcal{L}_\theta^w$ also maximizes the ELBO (Eq. 10) and thus the likelihood of $a_t$, so flowing toward a specific action makes it more likely. This intuition aligns naturally with the policy gradient objective: we want to increase the probability of high-advantage actions. By redirecting flow toward such actions (i.e., minimizing their diffusion loss), we make them more likely under the learned policy.

Using this relationship, we express the FPO ratio (Eq. 11) in terms of the flow matching objective:

$$r_\theta^{\mathrm{FPO}} = \frac{\exp(\mathrm{ELBO}_\theta(a_t|o_t))}{\exp(\mathrm{ELBO}_{\theta_{\mathrm{old}}}(a_t|o_t))} = \exp(\mathcal{L}_{\theta_{\mathrm{old}}}^w(a_t) - \mathcal{L}_\theta^w(a_t)), \tag{16}$$

where $\mathcal{L}_\theta^w$, as per Equation 7, can be estimated by averaging over $N_{\mathrm{mc}}$ draws of $(\tau, \epsilon)$. We find the sample count $N_{\mathrm{mc}}$ to be a useful hyperparameter for controlling learning efficiency. This estimator recovers the exact FPO ratio in the limit, although we use only a few draws in practice.

One possible concern with smaller $N_{\mathrm{mc}}$ values is bias. A ratio estimated from only one $(\tau, \epsilon)$ pair,

$$\hat{r}_\theta^{\mathrm{FPO}}(\tau, \epsilon) = \exp(\ell_{\theta_{\mathrm{old}}}^w(\tau, \epsilon) - \ell_\theta^w(\tau, \epsilon)), \tag{17}$$

is in expectation only an upper-bound of the true ratio. This can be shown by Jensen's inequality:

$$\mathbb{E}_{\tau, \epsilon}[\hat{r}_\theta^{\mathrm{FPO}}(\tau, \epsilon)] \geq r_\theta^{\mathrm{FPO}}. \tag{18}$$

To understand the upward bias, we can use the log-derivative trick to decompose the FPO gradient:

$$\nabla_\theta \hat{r}_\theta^{\mathrm{FPO}}(\tau, \epsilon) = -\hat{r}_\theta^{\mathrm{FPO}}(\tau, \epsilon)\nabla_\theta \ell_\theta^w(\tau, \epsilon). \tag{19}$$

Since the gradient operator commutes with expectation, the gradient term on the right side is unbiased:

$$\mathbb{E}_{\tau, \epsilon}[-\nabla_\theta \ell_\theta^w(\tau, \epsilon)] = -\nabla_\theta \mathcal{L}_\theta^w(a_t) = \nabla_\theta \mathrm{ELBO}_\theta(a_t). \tag{20}$$

In other words, gradient estimates are directionally unbiased even with worst-case overestimation of ratios. Our experiments are consistent with this result: while additional samples help, we observe empirically in Section 4.2 that FPO can be trained to outperform Gaussian PPO even with $N_{\mathrm{mc}} = 1$. Algorithm 1 details FPO's practical implementation using this mathematical framework.

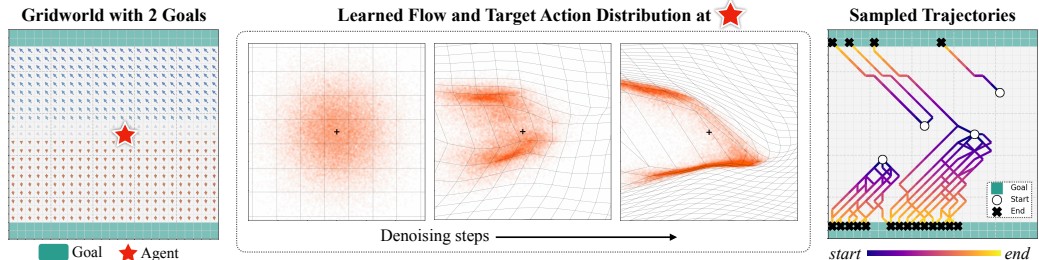

Figure 1: **Grid World**. (Left) 25×25 GridWorld with green goal cells. Each arrow shows a denoised action sampled from the FPO-trained policy, conditioned on a different latent noise vector. (Center) At the saddle-point state ($\star$) shown on the left, we visualize three denoising steps $\tau$ as the initial Gaussian gradually transforms into the target distribution through the learned flow, illustrated by the deformation of the coordinate grid. (Right) Sampled trajectories from the same starting states reach different goals, illustrating the multimodal behavior captured by FPO.

## 3.5 DENOISING MDP COMPARISON

Existing algorithms (Black et al., 2023; Ren et al., 2024; Liu et al., 2025) for on-policy reinforcement learning with diffusion models reformulate the denoising process itself as a Markov Decision Process (MDP). These approaches bypass flow model likelihoods by instead treating every step in the sampling chain as its own action, each parameterized as a Gaussian policy step. This has a few limitations that FPO addresses.

First, denoising MDPs multiply the horizon length by the number of denoising steps (typically 10-50), which increases the difficulty of credit assignment. Second, these MDPs do not consider the initial noise sample during likelihood computation. Instead, these noise values are treated as observations from the environment—this significantly increases the dimensionality of the learning problem. Finally, denoising MDP methods are limited to stochastic sampling procedures by construction. Meanwhile, FPO inherits the flexibility of all flow/diffusion sampler choices, including fast deterministic and higher-order samplers. Perhaps most importantly, FPO is simpler to implement because it does not require a custom sampler or the notion of extra environment steps.

## 4 EXPERIMENTS

We assess FPO's effectiveness by evaluating it in multiple domains. Our experiments include: (1) an illustrative GridWorld environment using Gymnasium (Brockman et al., 2016; Towers et al., 2024), (2) continuous control tasks with MuJoCo Playground (Zakka et al., 2025; Todorov et al., 2012), and (3) physics-based humanoid control in Isaac Gym (Makoviychuk et al., 2021). These tasks vary in dimensionality, reward sparsity, horizon length, and simulation environments.

### 4.1 GRIDWORLD

We first test FPO on a 25×25 GridWorld environment designed to probe the policy's ability to capture multimodal action distributions. As shown in Figure 1 left, the environment consists of two high reward regions located as the top and bottom of the map (green cells). The reward is sparse: agents receive a single reward upon reaching a goal or a penalty, with no intermediate rewards. This setup creates saddle points where multiple distinct actions can lead to equally successful outcomes, offering a natural opportunity to model diverse behaviors.

We train a diffusion policy from scratch using FPO by modifying a standard implementation (Yu, 2020) of PPO. The policy is parameterized as a two-layer MLP modeling $p(a_t \mid s, a_t^\tau)$, where $a_t \in \mathbb{R}^2$ is the action, $s \in \mathbb{R}^2$ is the grid state, and $a_t^\tau \in \mathbb{R}^2$ is the latent noise vector at noise level $\tau$, initialized from $\mathcal{N}(0, I)$ at $\tau = 0$. FPO consistently maximizes the return in this environment. The arrows in Figure 1 left shows denoised actions at each grid location, computed by conditioning on a random $a_t^\tau \sim \mathcal{N}(0, I)$ and running 10 steps of Euler integration. In Figure 1 center, we probe the learned policy by visualizing the flow over its denoising steps at the saddle point illustrated by

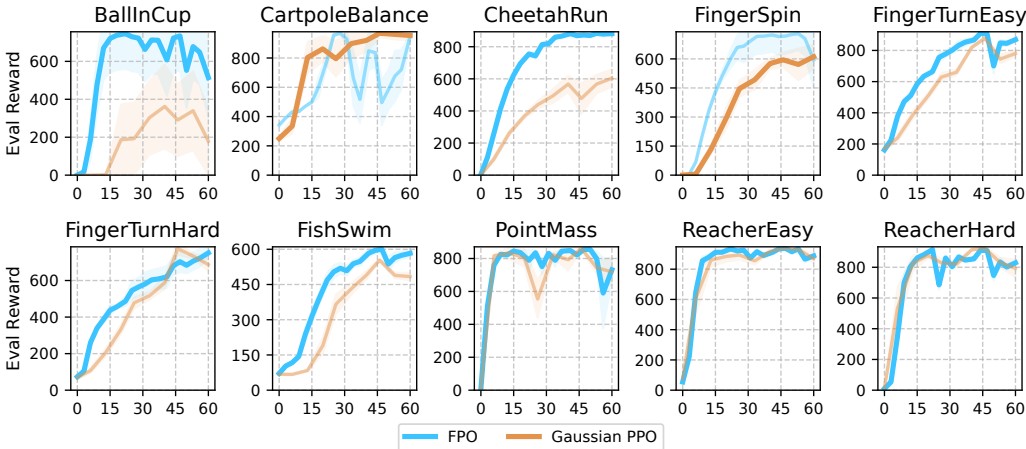

Figure 2: **Comparison between FPO and Gaussian PPO (Schulman et al., 2017) on DM Control Suite tasks.** Results show evaluation reward mean and standard error (y-axis) over 60M environment steps (x-axis). We run 5 seeds for each task; the curve with the highest terminal evaluation reward is bolded.

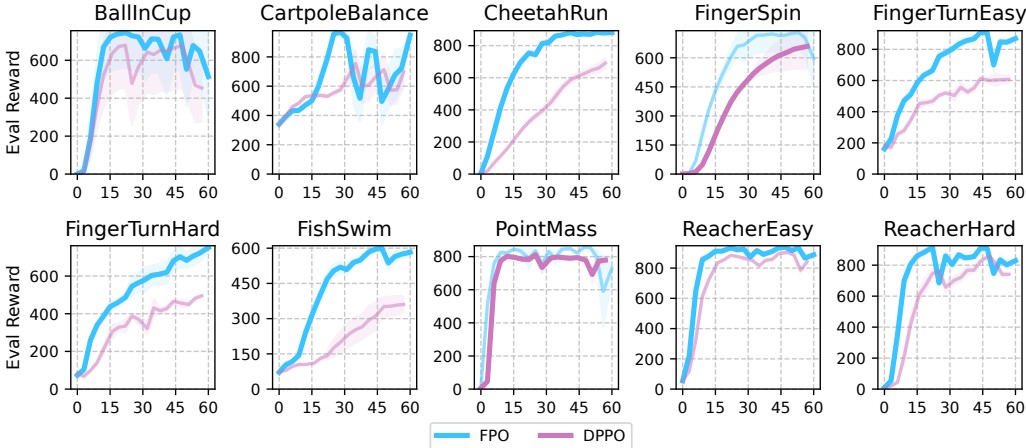

Figure 3: **Comparison between FPO and DPPO (Ren et al., 2024) on DM Control Suite tasks.** Results show evaluation reward mean and standard error (y-axis) over 60M environment steps (x-axis). We run 5 seeds for each task; the curve with the highest terminal evaluation reward is bolded.

the red star where either going up or down leads to the optimal reward. The initial Gaussian evolves into a bimodal distribution, demonstrating that the policy captures the multi-modality of the solution at this location. Figure 1 right shows multiple trajectories sampled from the policy, initialized from various fixed starting positions. The agent exhibits multimodal behavior, with trajectories from the same starting state reaching different goals. Even when heading toward the same goal, the paths vary significantly, reflecting the policy's ability to model diverse action sequences.

We also train a Gaussian policy using PPO, which successfully reaches the goal regions. Compared to FPO, it exhibits more deterministic behavior, consistently favoring the nearest goal with less variation in trajectory patterns. Please see the supplemental material for more details.

## 4.2 MUJOCO PLAYGROUND

Next, we evaluate FPO for continuous control using MuJoCo Playground (Zakka et al., 2025). We compare three policy learning algorithms: (i) a Gaussian policy trained using PPO, (ii) a diffusion policy trained using FPO, and (iii) a diffusion policy trained using DPPO (Ren et al., 2024). We evaluate these algorithms on 5 seeds for each of 10 environments adapted from the DeepMind Control Suite (Tassa et al., 2018; Tunyasuvunakool et al., 2020). Results are reported in Figures 2 and 3.

| Methods | Goal conditioning | Success rate (↑) | Alive duration (↑) | MPJPE (↓) |
|---|---|---|---|---|
| Gaussian PPO | All joints | **98.7**% | **200.46** | **31.62** |
| FPO | All joints | 96.4% | 198.00 | 41.98 |
| Gaussian PPO | Root + Hands | 46.5% | 142.50 | 97.65 |
| FPO | Root + Hands | **70.6**% | **171.32** | **62.91** |
| Gaussian PPO | Root | 29.8% | 114.06 | 123.70 |
| FPO | Root | **54.3**% | **152.90** | **73.55** |

Table 2: **Humanoid Control Quantitative Metrics.** We compare FPO with Gaussian PPO with different conditioning goals, and report the success rate, alive duration, and MPJPE averaged over all motion sequences.

**Policy implementations.** For the Gaussian policy baseline, we run the Brax (Freeman et al., 2021)-based implementation used by MuJoCo Playground (Zakka et al., 2025)'s PPO training scripts. We also use Brax PPO as a starting point for implementing both FPO and DPPO. Following Section 3.2, only small changes are required for FPO: noisy action and timestep inputs are included as input to the policy network, Gaussian sampling is replaced with flow sampling, and the PPO loss's likelihood ratio is replaced with the FPO ratio approximation. For DPPO, we make the same policy network modification, but apply stochastic sampling (Liu et al., 2025) during rollouts. We also augment each action in the experience buffer with the exact sampling path that was taken to reach it. Following the two-layer MDP formulation in DPPO (Ren et al., 2024), we then replace intractable action likelihoods with noise-conditioned sampling path likelihoods.

**Hyperparameters.** We match hyperparameters in Gaussian PPO, FPO, and DPPO training whenever possible: following the provided configurations in Playground (Zakka et al., 2025), all experiments use ADAM (Kingma, 2014), 60M total environment steps, batch size 1024, and 16 updates per batch. For FPO and DPPO, we use 10 sampling steps, set learning rates to 3e-4, and swept clipping epsilon $\varepsilon^{\text{clip}} \in \{0.01, 0.05, 0.1, 0.2, 0.3\}$. For DPPO, we perturb each denoising step with Gaussian noise with standard deviation $\sigma_t$, which we sweep $\in \{0.01, 0.05, 0.1\}$. We found that $\varepsilon^{\text{clip}} = 0.05$ produces the best FPO results and $\varepsilon^{\text{clip}} = 0.2, \sigma_t = 0.05$ produced the best DPPO results; we use these values for all experiments. For fairness, we also tuned learning rates and clipping epsilons for Gaussian PPO. We provide more details about hyperparameters and baseline tuning in the appendix.

**Results.** We observe in Figures 2 and 3 that FPO-optimized policies outperform both Gaussian PPO and DPPO, achieving higher rewards on 8 of 10 Playground tasks.

**Analysis.** In Table 1, we present average evaluation rewards for baselines, FPO, and several variations of FPO. We observe: **(1) $(\tau, \epsilon)$ sampling is important.** Decreasing the number of sampled pairs generally decreases evaluation rewards. More samples can improve learning without requiring more expensive environment steps. **(2) $\epsilon$-CFM is preferable over $u$-CFM in Playground.** $\epsilon$-CFM refers to computing conditional flow matching losses by first converting velocity estimates to $\epsilon$ noise values; $u$-CFM refers to CFM directly on velocity estimates. In Playground, we found that the former produces higher average rewards. We hypothesize that this is because $\epsilon$ scale is invariant to action scale, which results in better generalization for $\varepsilon^{\text{clip}}$ choices. For fairness, we also performed learning rate and clipping ratio sweeps for the $u$-MSE ablation. **(3) Clipping.** Like Gaussian PPO, the choice of $\varepsilon^{\text{clip}}$ in FPO significantly impacts performance.

| Method | Reward |
|---|---|
| Gaussian PPO | 667.8±66.0 |
| Gaussian PPO[†] | 577.2±74.4 |
| DPPO | 652.5±83.7 |
| FPO[‡] | **759.3±45.3** |
| FPO, 1 $(\tau, \epsilon)$ | 691.6±50.3 |
| FPO, 4 $(\tau, \epsilon)$ | 731.2±58.2 |
| FPO, $u$-MSE | 664.6±48.5 |
| FPO, $\varepsilon^{\text{clip}}$=0.1 | 623.3±76.3 |
| FPO, $\varepsilon^{\text{clip}}$=0.2 | 526.4±76.8 |

Table 1: **FPO variant comparison.** We report averages and standard errors across MuJoCo tasks. [†]Using default hyperparameters from MuJoCo Playground. [‡]FPO results use 8 $(\tau, \epsilon)$ pairs, $\epsilon$-MSE, $\varepsilon^{\text{clip}} = 0.05$.

## 4.3 HUMANOID CONTROL

Physics-aware humanoid control is higher-dimensional than standard MuJoCo benchmarks, making it a stringent test of FPO's capability. We therefore train a humanoid policy to track motion-capture (MoCap) trajectories in the PHC setting (Luo et al., 2023a) . This experiment follows the goal-

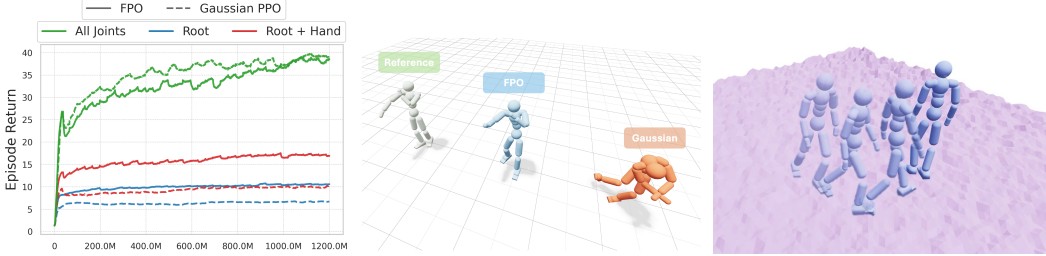

(a) Episode return along training.   (b) Root+hand conditioning.   (c) Rough terrain locomotion.

Figure 4: **Physics-based Humanoid Control.** (a) The curves show that FPO performance is close to that of Gaussian-PPO when conditioning on all joints and surpasses it when goals are reduced to the root or root+hands, indicating stronger robustness to sparse conditioning. (b) In the root+hands goal setting, FPO (blue) tracks the reference motion (grey) while Gaussian-PPO (orange) falls. (c) Trained with terrain randomization, FPO walks stably across procedurally generated rough ground.

conditioned imitation-learning paradigm pioneered by DeepMimic (Peng et al., 2018), in which simulated characters learn to reproduce reference motions. Sparse goals are significantly more challenging, requiring the policy to fill in the missing joint specification in a physically plausible manner.

**Implementation details.** Our simulated agent is a SMPL-based humanoid with 24 actuated joints, each with six degrees of freedom. The policy receives both proprioceptive observations and goal information computed from the motion-capture reference. A single policy is trained to track AMASS (Mahmood et al., 2019) motions following PHC (Luo et al., 2023a). We use the root height, joint positions, rotations, velocity, and angular velocity in a local coordinate frame as the robot state. For goal conditioning, we compute the difference between the tracking joint information (positions, rotations, velocity, and angular velocity) and the current robot's joint information, as well as the tracking joint locations and rotations. We explore both full conditioning, *i.e.,* conditioning on all joint targets, and under conditioning, *i.e.,* conditioning only on the root or the root and hands targets. Note that the same imitation reward based on all joints is used for both conditioning experiments. We adopt the per-joint tracking reward from DeepMimic (Peng et al., 2018).

**Evaluation.** For evaluation, we compute the success rate, considering an imitation unsuccessful if the average distance between the body joints and the reference motion exceeds 0.5 meters at any point. We also report the average duration the agent stays alive till it completes the tracking or falls. Finally, we compute the global mean per-joint position error (MPJPE) on the conditioned goals.

**Results.** Figure 4a shows that we successfully train FPO from scratch on this high-dimensional control task. With full joint conditioning, FPO performance is close to Gaussian PPO. However, when the model is under-conditioned—conditioned only on the root or the root and hands—FPO outperforms Gaussian PPO, highlighting the advantage of flow-based policies. While prior methods have achieved sparse-goal control with Gaussian policies, they first train a teacher policy that conditions on full joint reference and then distill the knowledge to sparse conditioned policies (Tessler et al., 2024; Li et al., 2025) or train a separate encoder observing sparse references (Luo et al., 2023b; 2024), highlighting the challenging nature of this problem setup.

Figure 4b visualizes the behaviors in the root+hands setting; FPO tracks the target closely, whereas the Gaussian policy drifts. Table 2 quantifies these trends, with FPO achieving higher success rates in the under-conditioned scenarios. Finally, as illustrated in Fig. 4c, FPO trained with terrain randomization enables the humanoid to traverse rough terrain, showing potential for sim-to-real transfer. Please see the supplemental for training hyperparameter details.

## 5 DISCUSSION

We introduce Flow Policy Optimization (FPO), a simple and intuitive algorithm for training flow-based generative models using policy gradients. We demonstrate FPO across a range of tasks where it shows promising results. Looking ahead, exciting future directions include investigating FPO on real robotic systems to test its sim-to-real capabilities. Another exciting direction is fine-tuning large pretrained diffusion policies (e.g., vision–language–action models) with FPO, and exploring how to incorporate one-step flow methods such as mean-flow Geng et al. (2025) for improved efficiency.

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

## A.1 APPENDIX

In this supplementary material, we discuss the deferred proofs of technical results, elaborate on the details of our experiments, and present additional visual results for the grid world, humanoid control, and present a text-to-image diffusion model finetuning experiment.

## A.2 FPO DERIVATION

The mathematical details presented in this section provide expanded derivations and additional context for the theoretical results outlined in Section 3 of the main text. Specifically, we elaborate on the connection between the conditional flow matching objective and the evidence lower bound (ELBO) first mentioned in Section 3.4, and provide complete derivations for the FPO ratio introduced in Section 3.3. These details are included for completeness and to situate our work within the theoretical framework established by Kingma et al. Kingma & Gao (2023), but are not necessary for understanding the core FPO algorithm or implementing it in practice.

First, we detail the different popular loss weightings used when training flow matching models laid out by Kingma et al. Kingma & Gao (2023). These weightings, denoted as $w(\lambda_t)$, determine how losses at different noise levels contribute to the overall objective and lead to different theoretical interpretations of Flow Policy Optimization.

Then, we show the more general result, which is that FPO optimizes the advantage-weighted expected ELBO of the noise-perturbed data. Specifically, for any monotonic weighting function (including Optimal Transport CFM schedules Lipman et al. (2023)), we can express the weighted loss as:

$$\mathcal{L}_\theta^w(a_t) = -\mathbb{E}_{p_w(\tau),q(a_t^\tau|a_t)}[\text{ELBO}_\tau(a_t^\tau)] + c_1, \tag{21}$$

where $p_w(\tau)$ is the distribution over timesteps induced by the weighting function, and $\text{ELBO}_\tau(a_t^\tau)$ is the evidence lower bound at noise level $\tau$ for the perturbed action $a_t^\tau$.

This means that FPO increases the likelihood of high-reward samples and the intermediate noisy samples $a_t^\tau$ from the sample path. By weighting this objective with advantages $\hat{A}_\tau$, we guide the policy to direct probability flow toward action neighborhoods that produce higher reward.

For diffusion schedules with uniform weighting $w(\lambda_\tau) = 1$, we show a somewhat stronger theoretical result. In this special case, the weighted loss directly corresponds to maximizing the ELBO of clean actions:

$$-\text{ELBO}(a_t) = \frac{1}{2}\mathbb{E}_{\tau\sim U(0,1),\epsilon\sim\mathcal{N}(0,I)}\left[-\frac{d\lambda}{d\tau}\cdot\|\hat{\epsilon}_\theta(a_t^\tau;\lambda_\tau) - \epsilon\|_2^2\right] + c_2, \tag{22}$$

which is a more direct connection to maximum likelihood estimation.

### A.2.1 LOSS WEIGHTING CHOICES

Most popular instantiations of flow-based and diffusion models can be reparameterized in the weighted loss scheme proposed by Kingma et al. Kingma & Gao (2023). This unified framework expresses each version as an instance of a weighted denoising loss:

$$\mathcal{L}_\theta^w(x) = \frac{1}{2}\mathbb{E}_{\tau\sim U(0,1),\epsilon\sim\mathcal{N}(0,I)}[w(\lambda_\tau)\cdot -\frac{d\lambda}{d\tau}\cdot\|\hat{\epsilon}_\theta(a_t^\tau;\lambda_\tau) - \epsilon\|_2^2], \tag{23}$$

where $w(\lambda_\tau)$ is a time-dependent function that determines the relative importance of different noise levels.

For those with a loss weight that varies monotonically with noise timestep $\tau$, the aforementioned relationship between the weighted loss and expected ELBO holds. Specifically, when $w(\lambda_\tau)$ is monotonically increasing with $\tau$, Kingma et al. prove:

$$\mathcal{L}_\theta^w(a_t) = -\mathbb{E}_{p_w(\tau),q(a_t^\tau|a_t)}[\text{ELBO}_\tau(a_t^\tau)] + c_1, \tag{24}$$

where $c_1$ is a constant, and does not vary with model parameters.

These monotonic weightings include several popular schedules: (1) standard diffusion with uniform weighting $w(\lambda_\tau) = 1$ Ho et al. (2020), (2) optimal transport linear interpolation schedule Lipman et al. (2023), which yields $w(\lambda_\tau) = e^{-\lambda/2}$, and (3) velocity prediction (v-prediction) with cosine schedule Salimans & Ho (2022), which also yields $w(\lambda_\tau) = e^{-\lambda/2}$.

### A.2.2 FLOW MATCHING AS EXPECTED ELBO OPTIMIZATION

To derive FPO in the more general flow matching case, we begin with the standard policy gradient objective, but replace direct likelihood maximization with maximization of the ELBO for noise-perturbed data:

$$\max_\theta \mathbb{E}_{a_t \sim \pi_\theta(a_t|o_t)} \left[ \mathbb{E}_{p_w(\tau), q(a_t^\tau|a_t)}[\text{ELBO}_\tau(a_t^\tau)] \cdot \hat{A}_t \right], \tag{25}$$

where $t$ is temporal rollout time and $\tau$ is diffusion/flow noise timestep.

This formulation directly leverages the result from Kingma et al. Kingma & Gao (2023) that for monotonic weightings, the weighted denoising loss equals the negative expected ELBO of noise-perturbed data plus a constant:

$$\mathcal{L}_\theta^w(a_t) = -\mathbb{E}_{p_w(\tau), q(a_t^\tau|a_t)}[\text{ELBO}_\tau(a_t^\tau)] + c_1. \tag{26}$$

To apply this within a trust region approach similar to PPO, we need to define a ratio between the current and old policies. Since we are working with expected ELBOs, the appropriate ratio becomes:

$$r^{\text{FPO}}(\theta) = \frac{\exp(\mathbb{E}_{p_w(\tau), q(a_t^\tau|a_t)}[\text{ELBO}_\tau(a_t^\tau)]_\theta)}{\exp(\mathbb{E}_{p_w(\tau), q(a_t^\tau|a_t)}[\text{ELBO}_\tau(a_t^\tau)]_{\theta,\text{old}})} \tag{27}$$

This ratio represents the relative likelihood of actions and their noisy versions under the current policy compared to the old policy.

It is important to note that the constant $c_1$ in the ELBO equivalence depends only on the noise schedule endpoints $\lambda_{min}$ and $\lambda_{max}$, the data distribution, and the forward process, but not on the model parameter $\theta$. This is critical for our derivation. It ensures that within a single trust region data collection and training episode, this constant remains identical between the old policy $\theta_{old}$ and the updated policy $\theta$. Consequently, when forming the ratio $r^{\text{FPO}}(\theta)$, these constants cancel out:

$$r^{\text{FPO}}(\theta) = \frac{\exp(\mathbb{E}_{p_w(\tau), q(a_t^\tau|a_t)}[\text{ELBO}_\tau(a_t^\tau)]_\theta + c_1)}{\exp(\mathbb{E}_{p_w(\tau), q(a_t^\tau|a_t)}[\text{ELBO}_\tau(a_t^\tau)]_{\theta,\text{old}} + c_1)} = \frac{\exp(\mathbb{E}_{p_w(\tau), q(a_t^\tau|a_t)}[\text{ELBO}_\tau(a_t^\tau)]_\theta)}{\exp(\mathbb{E}_{p_w(\tau), q(a_t^\tau|a_t)}[\text{ELBO}_\tau(a_t^\tau)]_{\theta,\text{old}})} \tag{28}$$

We estimate this ratio through Monte Carlo sampling of timesteps $\tau$ and noise $\epsilon$:

$$\hat{r}^{\text{FPO}}(\tau, \epsilon) = \exp(-\ell_\theta(\tau, \epsilon) + \ell_{\theta,\text{old}}(\tau, \epsilon)), \tag{29}$$

where $\ell_\theta(\tau, \epsilon) = \frac{1}{2}[-\dot{\lambda}(\tau)]\|\hat{\epsilon}_\theta(a_t^\tau; \lambda_\tau) - \epsilon\|^2$ is the reparameterized conditional flow matching loss for a single draw of random variables $\epsilon$ and $\tau$.

As discussed in the main text, $\hat{r}^{\text{FPO}}$ overestimates the scale but unbiasedly estimates the direction of the gradient. We can reduce or eliminate the scale bias by drawing more samples of $\tau$ and $\epsilon$.

### A.2.3 FPO WITH DIFFUSION SCHEDULES

For the special case of standard diffusion schedules with uniform weighting $w(\lambda_t) = 1$, we can derive a stronger theoretical result connecting our optimization objective directly to the ELBO of clean (non-noised) data.

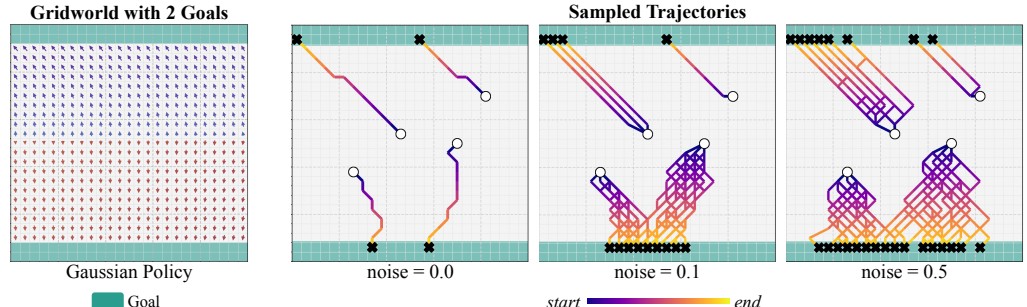

Figure A.1: **GridWorld with Gaussian Policy.** Left) $25 \times 25$ GridWorld with green goal cells. Each arrow shows an action predicted by the Gaussian policy. Right) Four rollouts under test-time noise perturbations ($\sigma = 0.0, 0.1, 0.5$). While the Gaussian policy achieves the goal, its trajectories lack diversity and hit the same goal consistently when given the same initialization point.

As shown by Kingma et al. Kingma & Gao (2023), when using uniform weighting, the weighted loss directly corresponds to the negative ELBO of the clean data plus a constant:

$$-\text{ELBO}(a_t) = \frac{1}{2}\mathbb{E}_{\tau \sim U(0,1), \epsilon \sim \mathcal{N}(0,I)}\left[-\frac{d\lambda}{d\tau} \cdot \|\hat{\epsilon}_\theta(a_t^\tau; \lambda_\tau) - \epsilon\|_2^2\right] + c_2, \tag{30}$$

where $c_2$ is a different constant than $c_1$ that also does not depend on model parameter $\theta$.

This means that minimizing the unweighted loss ($w(\lambda_\tau) = 1$) is equivalent to maximizing the ELBO of the clean action $a_t$, providing a more direct connection to traditional maximum likelihood estimation.

In the context of FPO, we can therefore express our advantage-weighted objective as:

$$\max_\theta \mathbb{E}_{a_t \sim \pi_\theta(a_t|o_t)}\left[\text{ELBO}_\theta(a_t) \cdot \hat{A}_t\right] \tag{31}$$

In this case, the objective direct increases a lower bound of the log-likelihood of clean actions $a_t$ weighted by their advantages, rather than over noise-perturbed actions.

The FPO ratio in this case becomes:

$$r^{\text{FPO}}(\theta) = \frac{\exp(\text{ELBO}_\theta(a_t))}{\exp(\text{ELBO}_{\theta,\text{old}}(a_t))} \tag{32}$$

This specific case highlights the close relationship between FPO and traditional maximum likelihood methods common for PPO Schulman et al. (2017). FPO still retains the computational advantages of avoiding explicit likelihood computations.

As in the general case, our Monte Carlo estimator exhibits upward bias of gradient scale. We can use the same PPO clipping mechanism to control the magnitude of parameter changes.

### A.2.4 ADVANTAGE-WEIGHED FLOW MATCHING DISCUSSION

Advantage estimates are typically zero-centered to reduce variance in estimating the policy gradient. Flow matching, however, learns probability flows which must be nonnegative by construction. Since advantages function as loss weights in this context, they should remain positive for mathematical consistency. A constant shift does not affect policy gradient optimization, which follows from the same baseline-invariance property that justifies using advantages in the first place. We find that both processed and unprocessed advantages work empirically.

## A.3 GRIDWORLD

Figure A.1 shows results from the Gaussian policy on the same Grid World trained using PPO. While the Gaussian policy can learn optimal behaviors, the trajectories resulting from it are not as diverse

| Learning Rate | Clipping Epsilon ($\varepsilon^{\text{clip}}$) | | | | | |
|---|---|---|---|---|---|---|
| | 0.3 | 0.2 | 0.1 | 0.05 | 0.03 | 0.01 |
| 0.0001 | 589.5 | 648.5 | 646.6 | 608.6 | 500.5 | 458.5 |
| 0.001 | 556.0 | 646.1 | 654.6 | 636.2 | 562.6 | 471.8 |
| 0.003 | 548.9 | 603.1 | 586.4 | 535.7 | 480.8 | 400.8 |
| 0.0003 | 567.0 | 631.8 | **667.8** | 650.9 | 570.4 | 492.0 |
| 0.0005 | 544.8 | 586.8 | 629.5 | 559.7 | 505.6 | 406.5 |

Table A.1: **Hyperparameter sweep for Gaussian PPO on the subset of Playground tasks that we evaluate on.** All quantities are average rewards across 10 tasks, with 5 seeds per task. The default configuration in Playground Zakka et al. (2025) (before tuning) uses learning rate 1e-3 and clipping epsilon 0.3; the tuned variant we use for results in the main paper body sets learning rate to 3e-4 and clipping epsilon to 0.1.

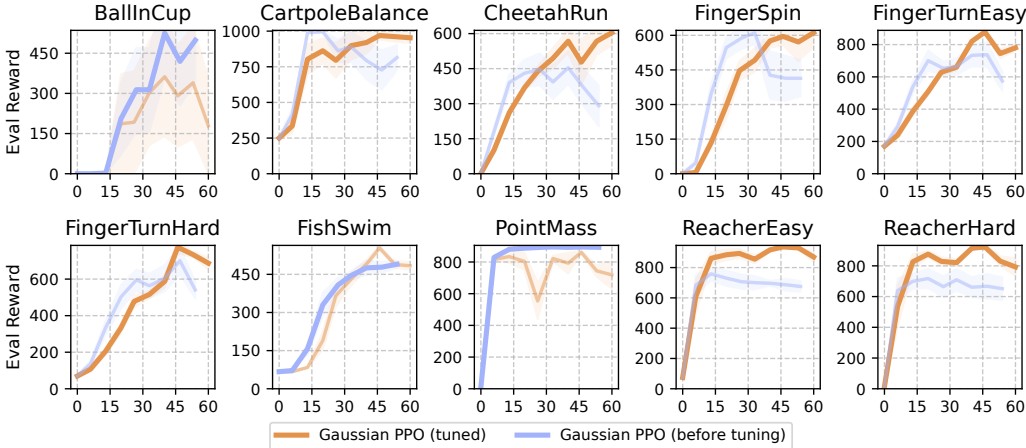

Figure A.2: **Gaussian PPO baseline results before and after tuning.** We tune clipping epsilon and learning rate to maximize average performance across tasks. Results show evaluation reward mean and standard error (y-axis) over 60M environment steps (x-axis). We run 5 seeds for each task; the curve with the highest terminal evaluation reward is bolded.

as those of the diffusion policy. We visualize 4 samples from the Gaussian policy with 0.0, 0.1, and 0.5 random noise perturbations at test time (Fig. A.1, right). Note that despite being initialized at the midpoint of the environment, all shown positions lead to a *single* goal mode, never both.

## A.4 MuJoCo Playground

Table A.2 shows hyperparameters used for PPO training in the MuJoCo Playground environment. These are imported directly from the configurations provided by MuJoCo Playground Zakka et al. (2025), but after sweeping hyperparameters to tune learning rate and clipping coefficients (Table A.1). We visualize improvements from this sweep in Figure A.2. Our flow matching and diffusion-based policies use the same hyperparameters, but adjust the clipping coefficient, turn off the entropy coefficient, and for DPPO Ren et al. (2024), introduce a stochastic sampling variance to account for the change in policy representation.

## A.5 Humanoid Control

In Table A.3, we report the detailed hyperparameters that we used for training both the Gaussian policy with PPO and the Diffusion policy with FPO in the humanoid control experiment. Note

| Hyperparameter | Value |
|---|---|
| Discount factor ($\gamma$) | 0.995 (most environments) |
| | 0.95 (BallInCup, FingerSpin) |
| GAE $\lambda$ | 0.95 |
| Value loss coefficient | 0.25 |
| Entropy coefficient | 0.01 |
| Reward scaling | 10.0 |
| Normalize advantage | True |
| Normalize observations | True |
| Action repeat | 1 |
| Unroll length | 30 |
| Batch size | 1024 |
| Number of minibatches | 32 |
| Number of updates per batch | 16 |
| Number of environments | 2048 |
| Number of evaluations | 10 |
| Number of timesteps | 60M |
| Policy network | MLP (4 hidden layers, 32 units) |
| Value network | MLP (5 hidden layers, 256 units) |
| Optimizer | Adam |

Table A.2: **PPO hyperparameters imported from MuJoCo playground Zakka et al. (2025).**

that we use the same set of hyperparameters for both policies. In our project webpage, we also provide videos showing qualitative comparisons between the Gaussian policy and ours on tracking an under-conditioned reference, and visual results of FPO on different terrains.

| Hyperparameter | Value | Hyperparameter | Value |
|---|---|---|---|
| *Policy Settings* | | | |
| Hidden size | 512 | Solver step size | 0.1 |
| Action perturbation std | 0.05 | Target KL divergence | None |
| Number of environments | 4096 | Normalize advantage | True |
| *Training Settings* | | | |
| Batch size | 131072 | Minibatch size | 32768 |
| Learning rate | 0.0001 | LR annealing | False |
| LR decay rate | 1.5e-4 | LR decay floor | 0.2 |
| Update epochs | 4 | L2 regularization coef. | 0.0 |
| GAE lambda | 0.2 | Discount factor ($\gamma$) | 0.98 |
| Clipping coefficient | 0.01 | Value function coefficient | 1.2 |
| Clip value loss | True | Value loss clip coefficient | 0.2 |
| Max gradient norm | 10.0 | Entropy coefficient | 0.0 |
| Discriminator coefficient | 5.0 | Bound coefficient | 10.0 |

Table A.3: **Policy training hyperparameters for humanoid control.**

## A.6 IMAGE REWARD FINE-TUNING

We explore fine-tuning a pre-trained image diffusion model on a non-differentiable task using the JPEG image compression gym proposed in DDPO (Black et al., 2023). We report this experiment as a negative result for FPO, due to the difficulty of fine-tuning diffusion models on their own output. Specifically, we find that repeatedly generating samples from a text-to-image diffusion model and training on them is highly unstable, even with manually-specified uniform advantages. We believe

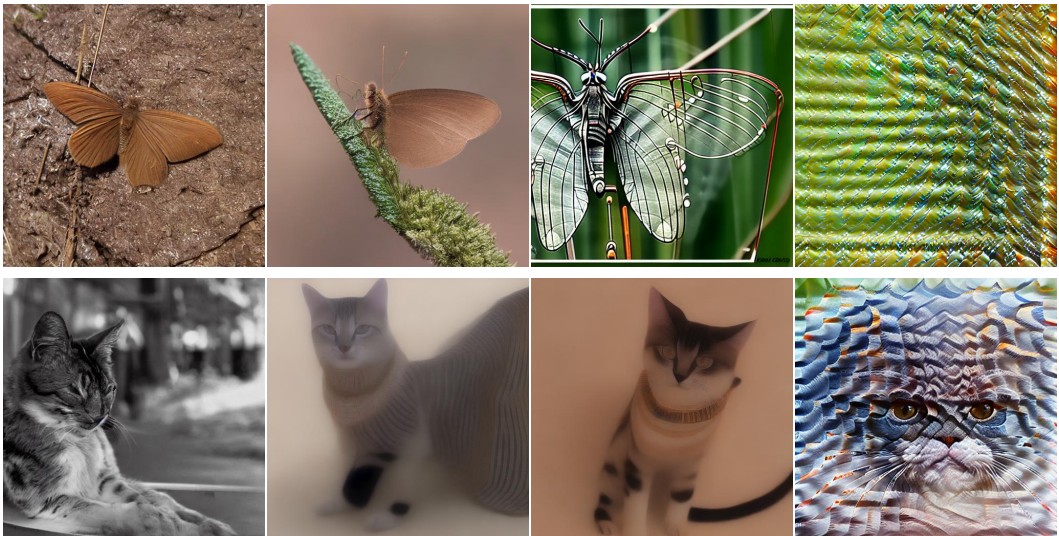

Figure A.3: **Image Generation at Different Training Steps.** We generate images using Stable Diffusion 1.5 finetuned with FPO as training progresses. We manually set all advantages to 1 to eliminate the reward signal and investigate the dynamics of sampling from a text-to-image diffusion model then training on the results in a loop. In the top row, we display images from a training run using a classifier-free guidance (CFG) scale of 4. In the bottom row, we display images from a training run using a CFG scale of 2. Low CFG scales tend to encourage bluriness while high CFG scales encourage saturation and sharp geometric artifacts. Both diverge after a few hundred epochs even with tuned hyperparameters.

that this is related to classifier-free guidance (CFG) Ho & Salimans (2022). CFG is necessary to generate realistic images, however it is sensitive to hyperparameters, where too much or too little guidance introduces artifacts such as blur or oversaturation that do not reflect the original training data. Sometimes these artifacts are not visible to human eyes. These artifacts are further amplified over successive iterations of RL epochs, ultimately dominating the training signal.

This phenomenon aligns with challenges previously identified in the literature on fine-tuning generative models on their own outputs (Shumailov et al., 2024; 2023; Alemohammad et al., 2024). To illustrate this, we fine-tune Stable Diffusion with all advantages set to 1 to eliminate the reward signal. This is equivalent to fine-tuning on self-generation data in an online manner. We explore CFG scales of 2 and 4 in Figure A.3. We find that both CFG scales induce a regression in quality over time. Specifically, the CFG scale of 2 makes the generation more blurry, while the scale of 2 causes the generated images to feature high saturation and geometry patterns. Both eventually diverge to abstract geometric patterns.

## A.7 LARGE LANGUAGE MODEL USE

We used Large Language Models (LLMs) to aid in polishing the writing of this paper. We also used LLM-based web agents to help discover relevant related works. No LLMs were used to generate original scientific content.

