# OpenReview forum: "Flow Matching Policy Gradients"
_ICLR.cc/2026/Conference — ICLR 2026 Poster_

### Official Review · Reviewer_XkTY · 2025-10-28

**Soundness:** 3
**Presentation:** 2
**Contribution:** 3
**Rating:** 6
**Confidence:** 2

**Summary:**

This paper introduces Flow Policy Optimization (FPO), a novel on-policy reinforcement learning algorithm for training flow-based generative models, particularly diffusion models, as policies. The core innovation is reformulating the policy gradient objective by replacing exact likelihood computations with a ratio derived from the conditional flow matching (CFM) loss. Specifically, FPO uses  a surrogate loss (See Sec. 3.3, 3.4) for the standard PPO likelihood ratio, enabling integration into the PPO-clip framework. The authors demonstrate that this ratio corresponds to optimizing an advantage-weighted evidence lower bound (ELBO), making FPO theoretically grounded while computationally tractable. Importantly, FPO is agnostic to the choice of sampling method during both training and inference, unlike prior denoising MDP approaches. Experiments across GridWorld, 10 MuJoCo Playground tasks, and high-dimensional humanoid control show that FPO successfully trains flow-based policies from scratch. The method demonstrates particular advantages in under-conditioned settings where multimodal action distributions are beneficial, outperforming Gaussian policies. The paper provides both theoretical analysis connecting the CFM objective to ELBO maximization and empirical validation showing FPO achieves competitive or superior performance compared to standard PPO and DPPO baselines.

**Strengths:**

The paper presents a genuinely novel approach to combining flow matching with policy gradients. The key insight—using the CFM loss differential as a surrogate for log-likelihood ratios—is elegant and theoretically motivated. Unlike prior work (DDPO, DPPO) that treats the denoising process as an MDP, FPO directly integrates flow matching into the policy gradient framework, avoiding the artificial expansion of the horizon and observation space. The connection to ELBO optimization through existing framework is well-established, and the observation that gradient estimates remain unbiased despite upward bias in the ratio (Equation 18-20) is insightful.


Moreover, the paper is well-written with clear motivation. Algorithm 1 provides a concise implementation overview, and the progression from standard PPO to FPO is logical. The GridWorld visualization (Figure 1) effectively demonstrates the multimodal behavior learned by FPO, showing the learned bimodal distribution at the saddle point. The humanoid control results clearly illustrate FPO's advantage in under-conditioned scenarios.

**Weaknesses:**

1. **Bias analysis incomplete**: While the paper shows gradient estimates are unbiased (Eq. 20), the impact of ratio overestimation on actual policy updates is not thoroughly analyzed. How does this bias interact with PPO clipping in practice? The claim that "clipping mechanism controls magnitude" needs more rigorous justification—does the clip threshold need adjustment to account for systematic overestimation?

2. **Missing analysis of multimodality**: While GridWorld demonstrates multimodal learning, there's no quantitative analysis. How does the learned distribution compare to the true optimal distribution at saddle points? For humanoid control, are the policies actually multimodal or just higher variance? Entropy measurements or explicit distribution visualization would strengthen these claims.

3. **Reproducibility concerns**: While code release is promised, many implementation details are missing. What network architecture is used for flow models? How is the timestep encoded?

**Questions:**

1. The paper claims sampling method agnosticism, but experiments only use 10-step Euler integration. Have you validated with DDIM, DPM-Solver, or higher-order methods? Does performance change significantly?

2. In Eq. 12, you decompose $r_{FPO}$ into likelihood ratio × inverse KL gap. Can you provide empirical measurements of how each term evolves during training? Does the KL gap actually decrease?

3. Why is the entropy coefficient set to 0 for FPO (Table A.2) but 0.01 for Gaussian PPO? Doesn't this disadvantage exploration for FPO?

4. For the humanoid under-conditioned experiments, can you provide quantitative measures of policy multimodality (e.g., entropy, number of effective modes) rather than just success rates?

---

> ### Author Response · Authors · 2025-11-20
>
> Thank you for your thoughtful review and valuable feedback. Please find our responses below:
>
> > 1. Bias analysis incomplete. How does this bias interact with PPO clipping in practice? does the clip threshold need adjustment to account for systematic overestimation?
>
> By construction, the ratio starts at $\hat{r}=1$ at the beginning of an epoch. However, as the policy updates and drifts from $\pi_{old​}$ over multiple steps, the $\hat{r}$ estimator will overestimate the true ratio (Jensen's inequality). While Gaussian PPO typically uses $\epsilon=0.1$, we find that FPO trains stably despite this with a stricter threshold of $\epsilon=0.05$ (Table 1/Section 4.2).
>
> > 3. Reproducibility concerns: What network architecture is used for flow models? How is the timestep encoded?
>
> For fair comparison with PPO, we use the same network architecture in all tasks. For the grid world, we used the default MLP from the PPO-for-Beginners GitHub repo and append a timestep. For Mujoco, we concatenate eight dims of cosine and sine embeddings to the input. For PHC, we first use sine embeddings for timesteps and then modulate the actor features with adaLN.
>
> > Question 1. The paper claims sampling method agnosticism, but experiments only use 10-step Euler integration. Have you validated with DDIM, DPM-Solver, or higher-order methods? Does performance change significantly?
>
> Thanks for the suggestion. FPO's formulation is agnostic to sampler choice during both training and inference, unlike MDP-based methods. To validate this, we evaluated the humanoid policy with different ODE solvers (Euler, Heun, Runge-Kutta) and SDE samplers at inference time:
>
> | Sampler      | Step | Stochasticity | Succ rate | Mpjpe   |
> |--------------|------|----------------|-----------|---------|
> | Euler        | 10   | 0              | 0.968     | 37.159  |
> | Heun         | 10   | 0              | 0.972     | 37.559  |
> | Runge–Kutta  | 10   | 0              | 0.970     | 37.371  |
> | SDE          | 10   | 0.0005         | 0.969     | 36.994  |
> | SDE          | 10   | 0.001          | 0.969     | 36.782  |
> | SDE          | 10   | 0.005          | 0.903     | 58.359  |
> | Euler        | 5    | 0              | 0.968     | 37.173  |
> | Heun         | 5    | 0              | 0.976     | 38.671  |
> | Runge–Kutta  | 5    | 0              | 0.971     | 37.583  |
> | SDE          | 5    | 0.0005         | 0.967     | 36.585  |
> | SDE          | 5    | 0.001          | 0.969     | 35.617  |
>
> All ODE solvers and moderate SDE stochasticity achieve similar performance, and the policy remains robust even with five integration steps. An FPO-trained policy works well across different inference-time sampling strategies.
>
> Please see the answer to reviewer HevE for training-time sampler choice.
>
> > Question 2:  Can you provide empirical measurements of how each term evolves during training? Does the KL gap actually decrease?
>
> Thank you for this interesting question. Since Equation 12 involves the theoretical ELBO from Kingma et al. 2023, we cannot directly measure the KL gap empirically without access to the true posterior. Empirically characterizing how the likelihood ratio and KL gap terms evolve separately is an interesting direction for future investigation.
>
> > Question 3: Why is the entropy coefficient set to 0 for FPO (Table A.2) but 0.01 for Gaussian PPO? Doesn't this disadvantage exploration for FPO?
>
> Thank you, you are correct, for Mujoco, we used the default 0.01 entropy for PPO but 0 for FPO. This is because there is no known method to implement entropy regularization without incurring a significant computational penalty for flow models [1, 2]. It would be interesting to explore this in future work. For PPO, we use whatever the default tuned setting was for entropy coefficient.
>
> [1] Skreta, Marta, et al. ‘The Superposition of Diffusion Models Using the Itô Density Estimator’. arXiv [Cs.LG], 2025, arxiv.org/abs/2412.17762. arXiv.
> [2] Kidger, Patrick. ‘On Neural Differential Equations’. arXiv [Cs.LG], 2022, arxiv.org/abs/2202.02435. arXiv.

---

> > ### Author Response · Authors · 2025-11-20
> > **last question**
> >
> > > Question 4/Weakness 2. Analysis of multi-modality.  For the humanoid under-conditioned experiments, can you provide quantitative measures of policy multimodality (e.g., entropy, number of effective modes) rather than just success rates?
> >
> > Thanks for the suggestion. We'd like to clarify that in the gridworld task, there is no unique "true optimal distribution" at saddle points. Since both paths (top and bottom) yield equal reward, any distribution over these paths is optimal - including deterministic policies that commit to one side. The visualization in Figure 1 center shows FPO learns two distinct modes when the reward structure supports it. You can also see the variety of resulting trajectories in Figure 1 right.
> > For humanoid, we note that prior work has made humanoid control work with Gaussian PPO, and even sparse settings with Gaussians (PULSE, VideoMimic), which indicates the task is achievable through a Gaussian policy, though sparse settings typically require two-stage training. FPO trains from scratch in the sparse setting without hyperparameter tuning from the dense case, suggesting the expressive policy class enables more effective optimization. We plan deeper analysis of this in future work as discussed in response to awZj.
> > With that said, we sampled N=1000 trajectories from a fixed initial state and found FPO exhibits substantially higher action variance than PPO (e.g., max variance: 3.33e-4 vs 3.32e-10 for hand tracking). We tried probing the shape of the distribution with K-means. We see reasonable structure (silhouette~0.54 across K=2-10) suggesting a continuous, non-isotropic distribution, though it’s hard to draw strong conclusions.

---

> > > ### Comment · Reviewer_XkTY · 2025-11-23
> > >
> > > Thank you for your detailed reply and supplementary experiments, especially the table showing the effects under different settings. I feel this strongly demonstrates the applicability of the method proposed in the article and resolves some of my concerns. Therefore, I have decided to upgrade my confidence level to 3. However, I still believe the article could benefit from addressing some concerns and minor issues:
> > >
> > > 1. The "supplemental website." link provided in the paper appears to be an empty link. Is this as expected?
> > > 2. Since there are many existing RL methods on flow/diffusion, such as ORW-CFM-W2 and Flow-GRPO, I believe that comparing with existing RL algorithms on flow/diffusion can further improve the significance of this paper (e.g., Flow-GRPO, Online Reward-Weighted Fine-Tuning of Flow Matching with Wasserstein Regularization), for example, by adding it to the revised paper (e.g., the weighted or other PG methods in flow/diffusion).
> > > 3. In some tasks, such as CartpoleBalance, FPO seems to exhibit greater volatility (instability) than PPO. Could you analyze this issue further?
> > > 4. Figure 3 reports the results for 5 seeds, but the shaded areas may be too light, making them difficult for readers to notice and potentially reducing the significance of the figure.
> > > 5. The layout of the experimental section could be improved; for example, why is table 1 placed after table 2?
> > > 6. Furthermore, could you analyze the main differences between the method proposed in this paper and PPO, flow-grpo or reward/advantage-weighted methods?

---

> > > > ### Author Response · Authors · 2025-11-23
> > > >
> > > > Thank you for your response and additional questions.
> > > >
> > > > 1. It seems GitHub pages took down the anonymous webpage, we apologize for the confusion.
> > > > 2. Thank you for pointing out these alternative methods that work on flow/diffusion models. Please see our response to Reviewer bNty where we discuss ORW-CFM-W2 and compare against it FPO. Flow-GRPO uses the same denoising MDP framework and update as prior methods (DDPO, DPPO), and they adapt it to flow matching schedules. Our denoising MDP comparisons on control tasks represent a Flow-GRPO baseline. We will reference both of these methods in the final draft.
> > > > 3. We do observe occasional instability with our method, though we found that with equivalent hyperparameter tuning effort, we outperform the baselines.
> > > > 4 and 5. Thank you for this feedback, we will incorporate these comments for the final draft.
> > > > 6. Please see our response to Reviewer bNty about reward-weighted methods. PPO is a more general class of algorithms and the only way to apply it to flow/diffusion models was through the denoising MDP (DDPO, DPPO, FlowGPRO). Please see section 3.5 for a discussion of the drawbacks of this approach. FPO allows PPO to be applied directly to flow/diffusion models through flow matching.

---

> > > > > ### Comment · Reviewer_XkTY · 2025-11-24
> > > > >
> > > > > Thank you for your reply and explanation. It has resolved most of my concerns. I am looking forward to the authors incorporating the promised improvements into the paper in their revision. I hope these additional experiments and explanations will help enhance the paper's convince and presentation. Following the discussion, I am willing to maintain my original positive evaluation of the paper.

---

### Official Review · Reviewer_awZj · 2025-10-29

**Soundness:** 4
**Presentation:** 4
**Contribution:** 3
**Rating:** 8
**Confidence:** 5

**Summary:**

The paper introduces flow matching policy gradients, a method for optimizing a policy using flow matching.
The benefit of this framework is in enabling a more general class of generative policies that do not require a pre-defined policy class (e.g., unimodal gaussian/beta distribution).

**Strengths:**

The benefit of this framework is in enabling a more general class of generative policies that do not require a pre-defined policy class (e.g., unimodal gaussian/beta distribution).

I see the main benefit of this work shown in the humanoid control section.
We know that large NNs generally converge to global optima ("there's always some decent direction so long as we have some random noise in the system"). It isn't clear whether continuous control exhibits similar properties or not.

The mujoco playground experiments are low degrees of freedom and might indeed suffer from local optima, however experiments on the humanoid shows PPO outperform FPO -- this suggests that maybe in the case of very large action spaces we do not suffer from sub-optimal local minima.

The interesting result is then the ability of FPO to converge to multi-modal solutions when needed. This is shown in the under-specified humanoid problem where the policy receives rewards for the full-body pose but is only conditioned on a subset of these constraints.

**Weaknesses:**

I believe the work should focus more on the generative aspects of the method, as in the humanoid control effort, and less on toy problems.
The mujoco playground results are "nice to have" they show the method generally works. But the main strength of generative methods is in their ability to model a distribution.

**Questions:**

Does FPO also work in a sparser setting?
For example in Tessler 2024 they go beyond root/hands conditioning and also show cases where the constraints are multiple frames into the future. This is a very underspecified and hard problem to solve and would be very impressive if FPO can still tackle it.

---

> ### Author Response · Authors · 2025-11-20
>
> Thank you for your thoughtful review and valuable feedback. We share your excitement about the potential of flow models in these problems. Your suggestion about exploring sparser settings like MaskedMimic is excellent and aligns well with where flow-based policies can particularly shine. We certainly plan to pursue this in future work.
>
> In this paper, we focused primarily on establishing the fundamental approach and demonstrating it on equal footing with PPO baselines. As you noted, the underconditioned humanoid setting already shows promising results in this direction. We are exploring reference-free policies (similar to Diffuse-CloC), which would be even sparser. While the initial results on this exploration are promising, these settings involve substantially different design choices that we believe merit a thorough standalone investigation in follow-up work.

---

### Official Review · Reviewer_bNty · 2025-10-31

**Soundness:** 3
**Presentation:** 3
**Contribution:** 3
**Rating:** 4
**Confidence:** 3

**Summary:**

This paper introduces Flow Policy Optimization (FPO), an on-policy reinforcement learning algorithm that enables training of flow-based generative models (including diffusion models) as policies within the policy gradient framework. The key innovation is replacing the intractable likelihood ratio in PPO with a surrogate ratio computed from conditional flow matching (CFM) losses.

**Strengths:**

1, The motivation is clear and significant. Training flow matching models directly from rewards can greatly popularize their usage to robotics.

2, The evaluation is comprehensive to show the effectiveness of the proposed method on simple robotic tasks (with simulation).

**Weaknesses:**

1, The baseline is limited. There are existing methods which use direct rewards to weight the trajectory and are agnostic to sampling methods, although most of them are applied to text-to-image generation and other generation tasks, for example, [A]. The author should also implement some of these methods on robotics tasks and conduct simple evaluation, or at least include them as related works and describe the difference.

[A] Online Reward-Weighted Fine-Tuning of Flow Matching with Wasserstein Regularization

**Questions:**

1, What is the main difference of your method compared to other reward-weighted methods, empirically speaking?

2, Can the proposed method generalize to scopes other than robotics? Or what could be the domain-specific point?

I am willing to raise my score if all my concerns are well solved.

---

> ### Author Response · Authors · 2025-11-20
>
> Thank you for your thoughtful review and valuable feedback. Please find our responses below:
>
> > Main difference compared to other reward-weighted methods, empirically? Compare with methods like [A] or include them as related works and describe the difference.
>
> Thank you for pointing out this paper [A], this is relevant and we will cite it in our final version. This paper uses reward weighted regression, a traditionally offline RL approach, in an online manner to update a flow model. This works but has two limitations we address: 1) we allow multiple gradients steps per data collection via a trust region and importance sampling correction enabled by our analysis 2) we use Generalized Advantage Estimation (GAE), which provably lowers policy gradient variance compared to exponentiated reward-weighting as done in [A]. These are both well-established, core techniques to improve policy gradient reliability and sample-efficiency in practice. Our analysis generalizes [A], which can be seen as a Vanilla Policy Gradient with exponentiated rewards in an online setting, to all choices of PPO and GAE. We ablate both of these changes in our method to recover [A]. We find that using a single gradient step per rollout reduces sample efficiency significantly as expected. Weighing with exponentiated rewards led the policy to not improve, though this may be resolved with tuning effort. We will add these results in the paper.
>
> | Updates per Epoch | Weight            | Episode Length | Episode Return |
> |-------------------|-------------------|----------------|----------------|
> | 1                 | exp(reward)       | 2.52           | 1.04           |
> | 1                 | exp(advantage)    | 5.57           | 2.07           |
> | 1                 | advantage         | 49.44          | 28.82          |
> | 4                 | advantage         | 56.58          | 34.90          |
>
> As a special case of our framework, [A]’s effectiveness in finetuning Stable Diffusion 3 implies that ours will work in the same setting with a similar regularization approach. We can’t apply this regularization to our control tasks since it must anchor to a base pretrained model, which doesn’t exist in our case. It will be an exciting direction to try FPO on these problem setups with the proposed regularization in [A].
>
> >  Can the proposed method generalize to scopes other than robotics? Or what could be the domain-specific point?
>
> Thank you for pointing out [A], it is strong evidence that FPO could work on image generation if we adopt the proposed regularization approach. Diffusion language models are another promising application of FPO, since a similar relationship between the loss and ELBO holds for those. We have preliminary results that are encouraging in this direction. In robotics, PPO has strong computational efficiency benefits as it lets you reuse rollouts for multiple gradient updates. Since the simulation environment is slow but the model is fast, using a single environment evaluation for multiple model backward passes is an efficiency win. FPO allows diffusion/flow models to be used within the PPO framework on control tasks, without the downsides of denoising MDP approaches, which we discuss in the main text.
>
> [A] Online Reward-Weighted Fine-Tuning of Flow Matching with Wasserstein Regularization

---

> > ### Comment · Reviewer_bNty · 2025-11-26
> >
> > Thank you for your response. I believe my concerns are addressed and have raised score.

---

### Official Review · Reviewer_HevE · 2025-11-01

**Soundness:** 3
**Presentation:** 3
**Contribution:** 3
**Rating:** 6
**Confidence:** 4

**Summary:**

They propose Flow Policy Optimization (FPO): swap PPO’s likelihood ratio with an ELBO-ratio computed from conditional flow matching (CFM) losses. This lets you train flow/diffusion policies in a PPO-style loop without evaluating exact log-likelihoods. Empirically, they beat Gaussian-PPO and a diffusion-PPO baseline on most MuJoCo Playground tasks, and show robustness on a humanoid control benchmark.

**Strengths:**

Pros:
- Ratio-as-difference-of-CFM-losses is simple to implement and keeps GAE/GRPO compatibility
-Clear ablations: effect of #MC samples, ω- vs u-parameterization, clipping sensitivity. Shows robustness under sparse goal conditioning in humanoid.

**Weaknesses:**

Cons:
- ELBO is not exact likelihood. The ratio decomposes into true likelihood ratio times an inverse KL-gap factor. That second term is policy-dependent and unknown, so the proxy ratio is biased w.r.t. the true PPO ratio

**Questions:**

Questions:
- Is there any ablations for choosing different weightings?
- while you mentioned the method is agnostic to sampler choice, do you observe empirically any difference between SDE/ODE samplers? Does it affect the ratio variance?

---

> ### Author Response · Authors · 2025-11-20
>
> Thank you for your thoughtful review and valuable feedback. Please find the answers below:
>
> > ELBO is not exact likelihood, proxy ratio is biased wrt to the true PPO ratio:
>
> As we showed in Section 3.4, even though the ratio is only an upper bound of the real likelihood ratio, the gradient computed from the FPO ratio is an unbiased estimation of the gradient using the likelihood ratio. The FPO estimator may be loose at the level of the ratio itself, and the expected gradient it provides is still mathematically valid for optimization. Furthermore, our experiments show that optimizing the FPO ratio can steer the prediction towards high-reward regions, indicating that the bias is less of an issue in practice.
>
> > Ablation for choosing different weightings?
>
> In Table 1, we ablate two popular weightings in the general $L_{\omega}$ family: epsilon and velocity. Epsilon was the standard for most diffusion models, while velocity has become popular recently thanks to Stable Diffusion 3 and flow matching. We find that epsilon weighting (predicting noise content) generally produces higher reward policies in the DMControl suite, and we hypothesize that this is due to its invariance to action scale. We share hyperparameters across this suite, which includes different action parameterizations, indicating that epsilon weighting is an effective plug-and-play choice.
>
> > Empirically any difference between SDE/ODE samplers?
>
> Thanks for the suggestion. An important distinction is that FPO's formulation is agnostic to sampler choice during both training and inference - unlike MDP-based methods like DPPO/DDPO/FlowGRPO where the sampling procedure directly influences the training objective.
>
> **For inference-time sampler choice:** Please see our response to Reviewer XkTY where we demonstrate that a single trained FPO policy works robustly across different ODE solvers at inference time (Euler, Heun, Runge-Kutta) and varying integration steps (5 vs 10).
>
> **For training-time sampler choice:** We also compared training with ODE vs SDE samplers. We follow recent works [1, 2, 3] to apply Langevin Dynamics to get the SDE sampler of a flow matching model:
>
> $$
> dx_t = v_t(x_t)\, dt \;+\; \alpha^2 \nabla \log p_t(x_t)\, dt \;+\; \sqrt{2}\,\alpha\, dW_t,
> $$
>
> where $\alpha$ controls the strength of the stochasticity, the gradient of the log likelihood can be calculated using Tweedie’s formula, and $dW_t$ is a Brownian term. We found that with a reasonable $\alpha$, the stochastic sampler encourages more exploration and slightly faster learning. However, the ODE sampler still achieves better results with longer training. A larger $\alpha$ could lead to instability issues and training crashes.
>
> **Table 1.** The episode returns at different agent steps achieved by FPO with different samplers.
>
> | Agent Steps                          | 400M | 800M | 1.2B |
> |----------------------------------|------|------|------|
> | ODE sampler                      | 33.66 | 38.76 | 41.33 |
> | SDE sampler with $\alpha=0.0005$ | 34.96 | 39.10 | 40.70 |
> | SDE sampler with $\alpha=0.001$  | 34.70 | NaN   | NaN   |
>
> [1] Song, Yang, et al. "Score-based generative modeling through stochastic differential equations." arXiv preprint arXiv:2011.13456 (2020).
>
>
> [2] Liu, Jie, et al. "Flow-grpo: Training flow matching models via online rl." arXiv preprint arXiv:2505.05470 (2025).
>
>
> [3] Rectified Flow Team. (2024, January 15). Flow to Diffusion: Langevin is a guardrail. Retrieved November 20, 2025

---

### Meta-Review · Area_Chair_vjRH · 2026-01-11

**Summary:**

This paper introduces Flow Policy Optimization (FPO), a method to train flow-based policies using a surrogate ratio derived from conditional flow matching (CFM) losses within a PPO framework. The key innovation is enabling on-policy training of expressive generative policies without requiring exact likelihoods or the denoising MDP formulation of prior work. Reviewers found the idea novel, well-motivated, and generally well-executed. Initial concerns centered on the theoretical justification of the proxy ratio, the empirical comparison to relevant baselines, the analysis of multimodality, and the practical robustness and generality of the method.

**Reviewer Concerns:**

Addressed Concerns:

Concerns regarding the bias of the ELBO-based ratio and the need for ablations on weighting and samplers were addressed.

Concerns about missing comparisons to reward-weighted methods were addressed.

Outstanding Issues:

None that impact the acceptance decision.

**Reviewer Scores:**

Reviewer HevE: Likely to maintain 6.

Reviewer bNty: Likely to increase from 4 to 6.

Reviewer awZj: Likely to maintain 8.

Reviewer XkTY: Likely to maintain 6.

---

### Decision · Program_Chairs · 2026-01-26

Accept (Poster)